# PolyConf: Unlocking Polymer Conformation Generation through Hierarchical Generative Models

**Fanmeng Wang** [1 2 †]  **Wentao Guo** [3]  **Qi Ou** [4]  **Hongshuai Wang** [2]
**Haitao Lin** [2 †]  **Hongteng Xu** [1 5 6]  **Zhifeng Gao** [2]

## Abstract

Polymer conformation generation is a critical task that enables atomic-level studies of diverse polymer materials. While significant advances have been made in designing conformation generation methods for small molecules and proteins, these methods struggle to generate polymer conformations due to their unique structural characteristics. Meanwhile, the scarcity of polymer conformation datasets further limits progress, making this important area largely unexplored. In this work, we propose PolyConf, a pioneering tailored polymer conformation generation method that leverages hierarchical generative models to unlock new possibilities. Specifically, we decompose the polymer conformation into a series of local conformations (i.e., the conformations of its repeating units), generating these local conformations through an autoregressive model, and then generating their orientation transformations via a diffusion model to assemble them into the complete polymer conformation. Moreover, we develop the first benchmark with a high-quality polymer conformation dataset derived from molecular dynamics simulations to boost related research in this area. The comprehensive evaluation demonstrates that PolyConf consistently outperforms existing conformation generation methods, thus facilitating advancements in polymer modeling and simulation. The whole work is available at https://polyconf-icml25.github.io.

[†]Work done during an internship at DP Technology [1]Gaoling School of Artificial Intelligence, Renmin University of China [2]DP Technology [3]California Institute of Technology [4]SINOPEC Research Institute of Petroleum Processing Co., Ltd. [5]Beijing Key Laboratory of Research on Large Models and Intelligent Governance [6]Engineering Research Center of Next-Generation Intelligent Search and Recommendation, MOE. Correspondence to: Hongteng Xu <hongtengxu@ruc.edu.cn>, Zhifeng Gao <gaozf@dp.tech>.

*Proceedings of the 42nd International Conference on Machine Learning*, Vancouver, Canada. PMLR 267, 2025. Copyright 2025 by the author(s).

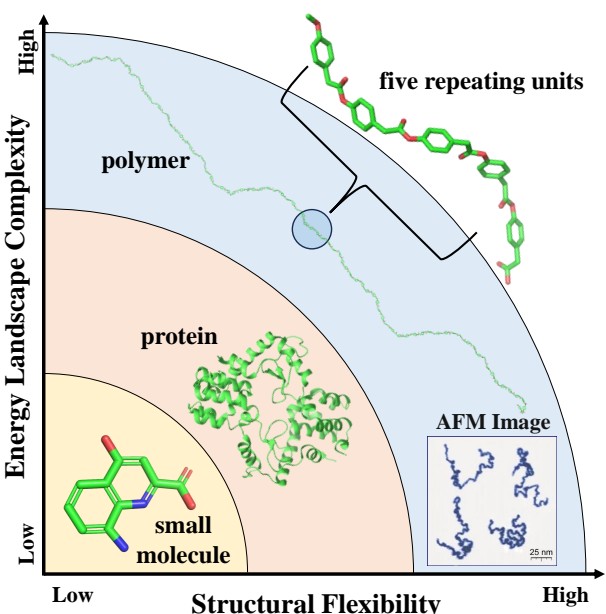

*Figure 1.* The comparison of small molecule, protein, and polymer. Here, the polymer chain comprises a series of repeating units, and the atomic force microscopy (AFM) image (Roiter & Minko, 2005) is used for direct 3D visualization of various polymer chains.

## 1. Introduction

Polymers, the macromolecules formed by covalent bonding of numerous identical or similar monomers, have already become indispensable to modern life (Audus & de Pablo, 2017; Kuenneth & Ramprasad, 2023). For example, polyethylene and polypropylene are used as durable and lightweight packaging materials, while polystyrene and polycarbonate find applications in electronics for their robustness and flexibility (Chen et al., 2021; Xu et al., 2023). The vast chemical space of polymers represents the art of molecular condensation that transforms simple building blocks into functional materials. In this context, polymer conformation generation [1], where various deep generative models can be used

---

[1]This work mainly focuses on generating conformations of linear homopolymers, as modeling copolymers and blends involves complexities beyond data modeling.

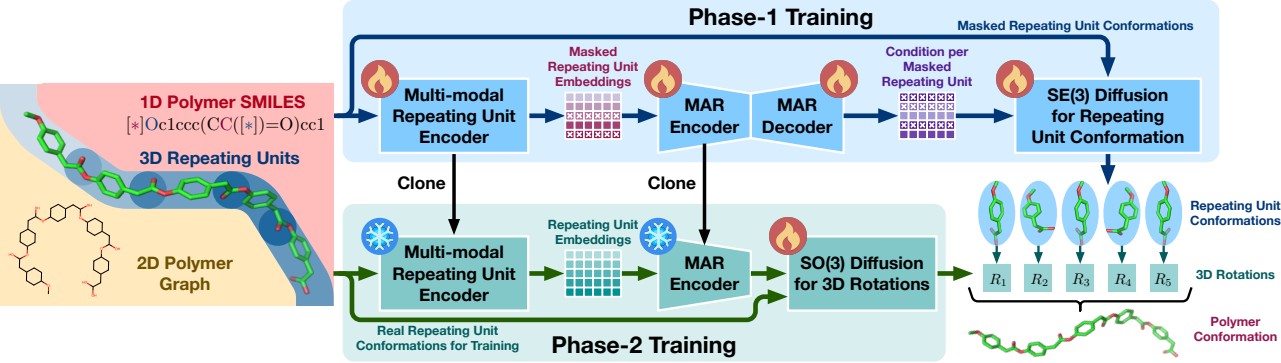

*Figure 2.* The overview of PolyConf: A hierarchical framework for polymer conformation generation, employing a masked autoregressive (MAR) model with a diffusion procedure to sample the conformation of each repeating unit within the polymer in random order, followed by a SO(3) diffusion model to assemble these repeating unit conformations into the complete polymer conformation. Here, for the sake of visualization simplicity, only a small fragment of the whole polymer conformation with five repeating units is presented in this figure.

to obtain the stable 3D polymer structures conditioned on the corresponding 2D polymer graph (Hoseini et al., 2021; Baillif et al., 2023), serves as a fundamental starting point for simulations—a crucial step in studying polymer properties (Kremer & Grest, 1990; Nakano & Okamoto, 2001).

Although conformation generation methods have been extensively studied over the past few years, generating macromolecular conformations remains challenging (Riniker & Landrum, 2015; Hawkins, 2017). Traditional stochastic methods like molecular dynamics (MD) simulations are widely used (Li et al., 2025a; Wang et al., 2025), but they are computationally expensive and slow, particularly for macromolecules like polymers (Pracht et al., 2020; Bilodeau et al., 2022), leading to the severe scarcity of polymer conformation datasets. Recent advances in artificial intelligence have led to various learning-based methods for conformation generation (Tang et al., 2024; Han et al., 2025). However, these methods are primarily designed for small molecules (Jing et al., 2022; Wang et al., 2024b) and proteins (Janson et al., 2023; Lu et al., 2024), leaving polymers unexplored (Kuenneth & Ramprasad, 2023; Wang et al., 2024a). As shown in Figure 1, polymers present unique challenges distinct from both small molecules and proteins. Unlike proteins, stabilized by strong, directional intramolecular interactions, polymers typically lack such organizing forces, resulting in more flexible and less ordered conformations. In addition, polymers occupy larger chemical space than small molecules, further complicating conformation generation (Martin & Audus, 2023). With fewer prior constraints, the complex conformational behaviors of polymers render protein conformation generation methods ineffective. Meanwhile, conformation generation methods designed for small molecules struggle to scale to polymers due to their significantly higher molecular weight (Xu et al., 2022). Given these challenges, it is critical to develop a tailored polymer conformation generation method to accommodate their unique structural

characteristics, thus fundamentally transforming our ability to design polymers and predict their properties.

Technically, a polymer conformation can be decomposed into a series of repeating unit conformations, as illustrated in Figure 1. However, unlike proteins that share a common backbone scaffold characterized by similarly distributed dihedral angles between amino acid residues (i.e., the "$N - C_\alpha - C - O$" structure) (Jumper et al., 2021), different polymers are composed of distinct monomers with diverse geometries, leading to various repeating units and structural frameworks (Huang et al., 2016). In addition, although repeating unit conformations within the same polymer share the same SMILES string and scaffold, the monomer itself can be highly complex and the spatial rearrangement of the remaining structure can also vary significantly in 3D space, which indicates that a polymer conformation cannot be simplistically modeled as a rigid assembly of a single predefined repeating unit conformation.

In this context, **we propose PolyConf, the first tailored method for polymer conformation generation**, to overcome the above challenges. As shown in Figure 2, we design our PolyConf as a hierarchical generative framework with a two-phase generating process. In the first phase, we leverage the state-of-the-art autoregressive generation paradigm proposed recently (Li et al., 2024), which integrates a masked autoregressive model (MAR) with a diffusion procedure (Ho et al., 2020), to generate repeating unit conformations in random order. Specifically, the partial output of the masked autoregressive model, which corresponds to masked repeating units, is used as the condition of the diffusion procedure designed for repeating unit conformation, thus effectively integrating the abilities of autoregressive modeling and diffusion processes to capture the dependencies within repeating unit conformations. In the second phase, building directly on the output of the MAR encoder, we employ an SO(3)

diffusion model [2] designed for repeating unit orientation transformation to generate the required orientation transformations, thus facilitating the assembly of repeating unit conformations into the complete polymer conformation.

Besides designing PolyConf, **we devote considerable time and resources to developing PolyBench, the first benchmark for polymer conformation generation,** to address the scarcity of polymer conformation datasets. PolyBench comprises a high-quality polymer conformation dataset obtained through molecular dynamics simulations (Afzal et al., 2020) and establishes standardized evaluation protocols. Extensive and comprehensive experiments on PolyBench consistently demonstrate that our PolyConf significantly outperforms existing conformation generation methods and achieves state-of-the-art performance, thus facilitating advancements in polymer modeling and simulation.

**The whole work, including code, model, and data, is publicly available** to facilitate the development of this important yet largely unexplored research area.

## 2. Related Work

### 2.1. Molecular Conformation Generation

In recent years, with the significant progress of deep generative models (Cao et al., 2024; Li et al., 2025b), many learning-based methods (Shi et al., 2021; Ganea et al., 2021) have been proposed to generate molecular conformations, thus facilitating traditional molecular dynamics simulations. In particular, CGCF (Xu et al., 2021) generates molecular conformations by combining the advantages of flow-based and energy-based models. Then GeoDiff (Xu et al., 2022) treats molecular conformations as point clouds and learns a diffusion model in Euclidean space, while TorsionalDiff (Jing et al., 2022) further restricts the diffusion process only in the torsion angle space to improve performance. Recently, MCF (Wang et al., 2024b) proposes to directly predict the 3D coordinates of atoms using the advantages of scale, and ETFlow (Hassan et al., 2024) tries to employ a well-designed flow matching to tackle this task. However, these molecular conformation generation methods are primarily designed for small molecules and typically generate the entire molecular conformation through a single process (Wang et al., 2023). Although they can be applied to generate polymer conformations, their performance will degrade significantly as polymers have larger chemical space and higher structural flexibility than small molecules.

### 2.2. Protein Conformation Generation

The emergence of AlphaFold (Jumper et al., 2021; Abramson et al., 2024) has greatly revolutionized the protein area (Bryant et al., 2022; Hekkelman et al., 2023) which inspired further protein conformation generation methods (Lu et al., 2024; Wang et al., 2024c) that unlock the multi-conformation capability by modifying AlphaFold predictions through MSA perturbations, such as mutations (Stein & Mchaourab, 2022), clustering (Wayment-Steele et al., 2024), and reducing depth (Del Alamo et al., 2022). In addition, various deep generative models, particularly diffusion models, have significantly contributed to protein conformation generation (Watson et al., 2022; Jing et al., 2023), including protein backbone generation (Huguet et al., 2024; Yim et al., 2024) and side chain generation (Zhang et al., 2024; Lee & Kim, 2024). Since proteins are composed of amino acid sequences with the consistent structural framework (i.e., the "$N - C_\alpha - C - O$" structure) (Yue et al., 2025), protein-specific prior knowledge, such as the unified backbone parameterization method (Yim et al., 2023; Bose et al., 2024) and constraints on the number of side-chain torsional angles (Misiura et al., 2022; Visani et al., 2024), has been widely integrated into various protein conformation generation methods. In this context, these methods are unsuitable for polymer conformation generation as they rely on evolutionary information, sequence similarity, and dihedral angle constraints that do not apply to polymers.

## 3. Proposed Method

### 3.1. Modeling Principle

**Frame-based Polymer Representation.** Each polymer with $N$ atoms can be represented as a 2D graph $\mathcal{G} = (\mathcal{V}, \mathcal{E})$, where $\mathcal{V} = \{v_i\}_{i=1}^N$ describes the corresponding atomic features (e.g., atomic type) and $\mathcal{E} = \{e_{ij}\}_{i,j=1}^N$ describes the corresponding bond features (e.g., bond type). The polymer conformation can be further denoted as $C = [c_i] \in \mathbb{R}^{N \times 3}$, where $c_i \in \mathbb{R}^3$ is the 3D coordinate of the $i$-th atom.

As shown in Figure 3, the decomposition of polymer conformation corresponds to a series of repeating unit conformations. Here, to model polymer structures effectively, we extend the standard definition of repeating units in polymer science, **incorporating the two key atoms from the neighboring repeating units (i.e., the atom-1 and atom-4 in Figure 3) into the conformation of the current repeating unit.** In this context, inspired by the modeling strategy of protein residue (Jumper et al., 2021), we further extract the frame from the corresponding repeating unit conformation. In particular, for the $i$-th repeating unit, its frame contains the corresponding orientation transformation [3], denoted as

---

[2] Since the corresponding repeating units in the polymer are defined by their monomers, the bonding atoms between adjacent repeating units are naturally overlapping (as illustrated in Figure 3). Therefore, we only need to train an SO(3) diffusion model to generate rotations, as translations can be directly derived from the 3D coordinates of those overlapping atoms.

---

[3] The corresponding orientation transformation is relative to the standard coordinate system.

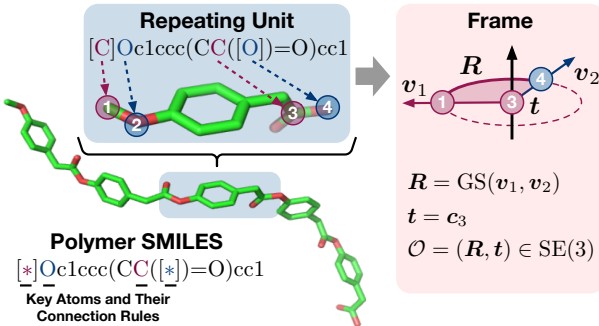

*Figure 3.* The illustration of frame-based polymer representation. In particular, the 1D polymer SMILES string represents the monomer's SMILES string with two "*" symbols marking polymerization sites. The 3D polymer conformation comprises a series of repeating unit conformations with identical SMILES strings but distinct 3D structures, overlapping at key atoms (e.g., atom-1 aligns with atom-3 of the previous repeating unit). The orientation transformation, derived from the key atoms within the corresponding frame, is denoted as $\mathcal{O} = (\boldsymbol{R}, \boldsymbol{t})$ where the rotation $\boldsymbol{R} \in \mathbb{R}^{3 \times 3}$ is calculated through the GramSchmidt operation (Leon et al., 2013) on vectors $\boldsymbol{v}_1$ and $\boldsymbol{v}_2$, and the translation $\boldsymbol{t} \in \mathbb{R}^3$ corresponds to the 3D coordinate of atom-3.

$\mathcal{O}_i = (\boldsymbol{R}_i, \boldsymbol{t}_i)$, where $\boldsymbol{R}_i \in \mathbb{R}^{3 \times 3}$ denotes the rotation and $\boldsymbol{t}_i \in \mathbb{R}^3$ denotes the translation. In this context, the polymer conformation can be further denoted as

$$\mathcal{C} = \{\mathcal{C}^u, \mathcal{O}\} = \{\{\boldsymbol{C}_i^u\}_{i=1}^{N_u}, \{\mathcal{O}_i\}_{i=1}^{N_u}\}, \quad (1)$$

where $\boldsymbol{C}_i^u = [\boldsymbol{c}_{i,j}^u] \in \mathbb{R}^{(\frac{N}{N_u}+2) \times 3}$ is the $i$-th repeating unit's conformation [4], $\mathcal{O}_i = (\boldsymbol{R}_i, \boldsymbol{t}_i)$ is the $i$-th repeating unit's orientation transformation, and $N_u$ is the number of repeating units in this polymer. For the $j$-th atom in the $i$-th repeating unit's conformation, its corresponding 3D coordinates in the polymer conformation can be expressed as $\boldsymbol{R}_i \boldsymbol{c}_{i,j}^u + \boldsymbol{t}_i$.

Please note that since each repeating unit involves 2 overlapping atoms from previous and next units, the number of atoms in one unit is not $\frac{N}{N_u}$ but rather $\frac{N}{N_u} + 2$.

**Hierarchical Generative Modeling.** The task of polymer conformation generation is a conditional generative problem, which aims to learn a generative model $p(\mathcal{C}|\mathcal{G})$ to model the empirical distribution of polymer conformations $\mathcal{C}$ (i.e., the stable 3D polymer structures) conditioned on the corresponding 2D polymer graph $\mathcal{G}$. Combined with Eq. (1), this generative model $p(\mathcal{C}|\mathcal{G})$ can be expressed as follows:

$$p(\mathcal{C}|\mathcal{G}) = p(\mathcal{C}^u, \mathcal{O}|\mathcal{G}) = p(\mathcal{C}^u|\mathcal{G}) \cdot p(\mathcal{O}|\mathcal{G}, \mathcal{C}^u), \quad (2)$$

where $\mathcal{C}^u = \{\boldsymbol{C}_i^u\}_{i=1}^{N_u}$ is the set of repeating unit conformations, and $\mathcal{O} = \{\mathcal{O}_i\}_{i=1}^{N_u}$ is the set of their corresponding orientation transformations.

[4]The repeating unit conformation is in the standard coordinate system through applying the inverse orientation transformation to the corresponding sub-structure within the polymer conformation.

Therefore, the polymer conformation generation task can be naturally denoted as a hierarchical generative process: (1) generating repeating unit conformations $\mathcal{C}^u$ based on the corresponding 2D polymer graph $\mathcal{G}$, i.e., $p(\mathcal{C}^u|\mathcal{G})$, and (2) then generating corresponding orientation transformations $\mathcal{O}$ of these repeating units given $\mathcal{G}$ and $\mathcal{C}^u$, i.e., $p(\mathcal{O}|\mathcal{G}, \mathcal{C}^u)$.

This hierarchical generative process further leads to the proposed PolyConf model and its two-phase learning strategy, as illustrated in Figure 2. In the following subsections, we will introduce them in detail.

### 3.2. Phase 1: Repeating Unit Conformation Generation

In this first phase, we employ the masked autoregressive model (Li et al., 2024) with a diffusion procedure to generate repeating unit conformations in random order, capturing their complex interactions rather than simple sequential dependencies. The distribution $p(\mathcal{C}^u|\mathcal{G})$ in Eq. (2) is therefore rewritten as follows:

$$\begin{aligned} p(\mathcal{C}^u|\mathcal{G}) &= p(\{\boldsymbol{C}_i^u\}_{i=1}^{N_u}|\mathcal{G}) \\ &= p(\{\mathcal{C}_k^u\}_{k=1}^K|\mathcal{G}) \\ &= \prod_{k=1}^K p(\mathcal{C}_k^u|\mathcal{G}, \{\mathcal{C}_i^u\}_{i=1}^{k-1}), \end{aligned} \quad (3)$$

where $\mathcal{C}^u = \{\boldsymbol{C}_i^u\}_{i=1}^{N_u}$ is the set of repeating unit conformations, and $\mathcal{C}_k^u$ is the corresponding subset of $\mathcal{C}^u$ that contains $\frac{N_u}{K}$ repeating unit conformations generated at the $k$-th step. Since these repeating unit conformations exist in continuous 3D space, we generate them in random order. Here, we define a random permutation $\pi$ to model this random order, and $\mathcal{C}_k^u$ can be expressed as:

$$\mathcal{C}_k^u = \{\boldsymbol{C}_{\pi(i)}^u \mid i \in \{(k-1)m+1, \ldots, km\}\}, \quad (4)$$

where $\boldsymbol{C}_{\pi(i)}^u$ is the corresponding conformation of the $\pi(i)$-th repeating unit, $m = \frac{N_u}{K}$ is the size of the subset, and $\pi$ ensures a random sampling order.

The key modules shown in Figure 2 are introduced below.

**Multi-modal Repeating Unit Encoder.** The multi-modal repeating unit encoder $\mathcal{M}$ comprises two parts: the 2D encoder $\mathcal{M}^{2d}$ for the whole polymer graph $\mathcal{G}$, and the 3D encoder $\mathcal{M}^{3d}$ for each repeating unit conformation $\boldsymbol{C}_i^u$ in the polymer conformation. The embedding extraction process can be expressed as follows:

$$\begin{aligned} \boldsymbol{X}^u &= \mathcal{M}(\mathcal{G}, \{\boldsymbol{C}_i^u\}_{i=1}^{N_u}) \\ &= \text{Concat}_1(\mathcal{M}^{2d}(\mathcal{G}), \text{Concat}_0(\{\mathcal{M}^{3d}(\boldsymbol{C}_i^u)\}_{i=1}^{N_u})) \\ &= \text{Concat}_1(\boldsymbol{X}^{2d}, \text{Concat}_0(\{\boldsymbol{X}_i^{3d}\}_{i=1}^{N_u})), \end{aligned} \quad (5)$$

where $\boldsymbol{X}^u \in \mathbb{R}^{N_u \times D_u}$ is the multi-modal repeating unit embeddings, $\boldsymbol{X}^{2d} \in \mathbb{R}^{N_u \times D_{2d}}$ is the 2D embedding of the whole polymer graph $\mathcal{G}$, $\boldsymbol{X}_i^{3d} \in \mathbb{R}^{1 \times D_{3d}}$ is the 3D embedding of the $i$-th repeating unit conformation $\boldsymbol{C}_i^u$, and

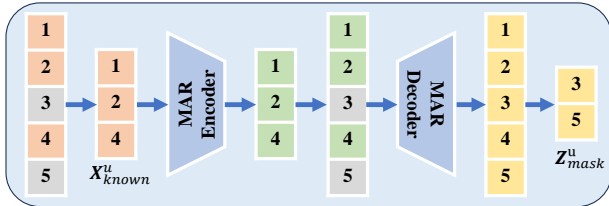

*Figure 4.* The illustration of the masked autoregressive modeling in the second phase, where grey blocks represent the corresponding embeddings of masked repeating units.

$\text{Concat}_i(\cdot)$ represents the corresponding concatenation operator in the $i$-th dimension.

In this work, our PolyConf adopts the encoder architecture from MolCLR (Wang et al., 2022) as its 2D encoder $\mathcal{M}^{2d}$ and the Uni-Mol (Zhou et al., 2023) as its SE(3)-invariant 3D encoder $\mathcal{M}^{3d}$. Besides, the entire multi-modal repeating unit encoder $\mathcal{M}$ is trainable in the first phase and remains frozen in the second phase.

**Masked Autoregressive Modeling.** As shown in Figure 2, to generate a random subset of unknown repeating unit conformations based on known/predicted repeating unit conformations iteratively (i.e., Eq. (3)), we employ the masked autoregressive modeling in the latent space of the multi-modal repeating unit encoder.

During training, given the random permutation $\pi$ and multi-modal repeating unit embeddings $\boldsymbol{X}^u \in \mathbb{R}^{N_u \times D_u}$, we randomly sample a masking ratio $\tau$ from the range $[0, 1]$, and then mask the corresponding repeating units, i.e.,

$$\boldsymbol{X}^u_{\text{known}} = \text{Concat}_0(\{\boldsymbol{X}^u_{\pi(i)} \mid i \in [(\tau N_u + 1), N_u]\}), \quad (6)$$

where $\pi(i)$ is the permuted repeating unit index used in Eq. (4), $\boldsymbol{X}^u_{\text{known}} \in \mathbb{R}^{(N_u - \tau N_u) \times D_u}$ is the embeddings of unmasked repeating units with known conformations, and $\boldsymbol{X}^u_{\pi(i)} \in \mathbb{R}^{1 \times D_u}$ refers to the $\pi(i)$-th row of multi-modal repeating unit embeddings $\boldsymbol{X}^u$ obtained through Eq. (5).

Furthermore, as illustrated in Figure 4, we use the MAR encoder $\Phi$ to encode $\boldsymbol{X}^u_{\text{known}}$ and then use the MAR decoder $\Psi$ to obtain the corresponding representations $\boldsymbol{Z}^u_{\text{mask}}$ of masked repeating units, i.e.,

$$\boldsymbol{Z}^u_{\text{mask}} = \Psi(\Phi(\boldsymbol{X}^u_{\text{known}})), \quad (7)$$

where $\boldsymbol{Z}^u_{\text{mask}} \in \mathbb{R}^{\tau N_u \times D_m}$ is the decoded representations of masked repeating units, the MAR encoder $\Phi$ and MAR decoder $\Psi$ are the standard Transformer architecture with the bidirectional attention mechanism.

**Diffusion Loss.** The goal of the masked autoregressive modeling is to generate repeating unit conformations based on the probability distribution of masked repeating unit conformations $\mathcal{C}^u_{\text{mask}}$ conditioned on the corresponding decoded

representations $\boldsymbol{Z}^u_{\text{mask}}$. As shown in Figure 3, repeating unit conformations are a specialized form of molecular conformations, characterized by the added complexity of interactions between repeating units. Given the recent success of diffusion models in generating molecular conformations, leveraging a diffusion model to represent this conditional probability distribution for each repeating unit is highly suitable. Following previous works (Xu et al., 2022; Li et al., 2024), the corresponding loss function can be formulated as a denoising criterion, i.e.,

$$\mathcal{L}_{\text{phase-1}} = \mathbb{E}_{\varepsilon,t}\left[\|\varepsilon - \varepsilon_\theta(\boldsymbol{C}^u_t | t, \boldsymbol{z}^u)\|^2\right],$$
$$\text{with } \boldsymbol{C}^u_t = \sqrt{\bar{\alpha}_t}\boldsymbol{C}^u + \sqrt{1 - \bar{\alpha}_t}\varepsilon, \quad (8)$$

where $\boldsymbol{C}^u \in \mathbb{R}^{(\frac{N}{N_u}+2) \times 3}$ is the conformation of one masked repeating unit (i.e., one element in $\mathcal{C}^u_{\text{mask}}$), $\boldsymbol{z}^u \in \mathbb{R}^{D_m}$ is the corresponding decoded representation of this masked repeating unit (i.e., the corresponding row of $\boldsymbol{Z}^u_{\text{mask}}$), $\bar{\alpha}_t$ is the predefined noise schedule, $t$ is the time step of this predefined noise schedule, $\varepsilon$ is the noise sampled from the predefined prior distribution, and $\varepsilon_\theta$ is the parameterized denoising network for noise estimator.

Specifically, we employ the diffusion process defined in the torsion angle space to model this probability distribution, and adopt the corresponding diffusion model architecture used in (Jing et al., 2022) as the denoising network $\varepsilon_\theta$.

### 3.3. Phase 2: Orientation Transformation Generation

As expressed in Eq (2), after generating repeating unit conformations $\mathcal{C}^u$ conditioned on the corresponding polymer graph $\mathcal{G}$ in the first phase, we still need to generate the corresponding orientation transformations $\mathcal{O}$ of these repeating units based on $\mathcal{G}$ and $\mathcal{C}^u$ in the second phase.

In particular, as illustrated in Figure 3, repeating unit conformations within the polymer partially overlap (e.g., the atom-1 of the current repeating unit aligns with the atom-3 of the previous repeating unit). Therefore, for each repeating unit's orientation transformation, i.e., $\mathcal{O}_i = (\boldsymbol{R}_i, \boldsymbol{t}_i)$, we only need to consider the generation of rotation $\boldsymbol{R}_i$ as the corresponding translation $\boldsymbol{t}_i$ can be directly derived by aligning those overlapping atoms after applying rotation $\boldsymbol{R}_i$.

**SO(3) Diffusion for Rotation Generation.** According to the above analysis, we can further simplify this problem to generating $\boldsymbol{R}_i$ for each repeating unit, i.e., modeling $p(\boldsymbol{R}|\mathcal{G}, \mathcal{C}^u)$, where $\mathbf{R} = [\boldsymbol{R}_i] \in \mathbb{R}^{N_u \times 3 \times 3}$. In this work, we develop an SO(3) diffusion model to generate $\mathbf{R}$, i.e.,

$$\widehat{\mathbf{R}}^{(0)} = \varphi(\mathcal{O}^{(t)}, t, \boldsymbol{E}^u), \text{ with } \mathcal{O}^{(t)} = (\mathbf{R}^{(t)}, \mathbf{T}^{(t)}). \quad (9)$$

Here, $\varphi$ denotes a denoising network, whose architecture is the same as the model used in (Yim et al., 2023). $\boldsymbol{E}^u \in \mathbb{R}^{N_u \times D_e}$ is the output of the MAR encoder (i.e., the condition concerning $\mathcal{G}$ and $\mathcal{C}^u$). $\mathbf{R}^{(t)} = [\boldsymbol{R}_i^{(t)}] \in \mathbb{R}^{N_u \times 3 \times 3}$ is

obtained through forward diffusion processes on $SO(3)^{N_u}$. $\mathbf{T}^{(t)} = [\boldsymbol{t}_i^{(t)}] \in \mathbb{R}^{N_u \times 3}$ is the translations calculated by aligning the overlapping atoms after applying the rotation transformations $\mathbf{R}^{(t)}$ to the repeating units.

The denoising network $\varphi$ takes the timestamp $t$, the transformations at time $t$ (i.e., $\mathcal{O}^{(t)} = (\mathbf{R}^{(t)}, \mathbf{T}^{(t)})$), and the condition information $\boldsymbol{E}^u$ as input, predicting rotation transformations for the repeating units, denoted as $\widehat{\mathbf{R}}^{(0)} = [\widehat{\boldsymbol{R}}_i^{(0)}] \in \mathbb{R}^{N_u \times 3 \times 3}$. Accordingly, we can learn it through minimizing the following loss function:

$$\mathcal{L}_{\text{phase-2}} = \frac{1}{N_u} \sum_{i=1}^{N_u} \|\widehat{\boldsymbol{R}}_i^{(0)} - \boldsymbol{R}_i\|^2, \qquad (10)$$

where $\boldsymbol{R}_i$ is the ground-truth rotation of the $i$-th unit.

### 3.4. Assembling for Polymer Conformation Generation

After generating the repeating unit conformations $\{\widehat{\boldsymbol{C}}_i^u\}_{i=1}^{N_u}$ in the first phase and the corresponding rotation transformations $\{\widehat{\boldsymbol{R}}_i\}_{i=1}^{N_u}$ in the second phase, the last step is to assemble these repeating unit conformations into the complete polymer conformation.

As illustrated in Section 3.1, the orientation transformation is relative to the standard coordinate system, meaning that the generated rotation transformations $\{\widehat{\boldsymbol{R}}_i\}_{i=1}^{N_u}$ are also relative to the standard coordinate system. Therefore, we first transform each generated repeating unit conformation $\widehat{\boldsymbol{C}}_i^u$ back to the standard coordinate system, i.e.,

$$\widehat{\boldsymbol{C}}_i^{u,\text{std}} = (\mathcal{O}_i^c)^{-1} \cdot \widehat{\boldsymbol{C}}_i^u = (\widehat{\boldsymbol{C}}_i^u - \boldsymbol{t}_i^c) \cdot (\boldsymbol{R}_i^c)^{-1}, \quad (11)$$

where $\mathcal{O}_i^c = (\boldsymbol{R}_i^c, \boldsymbol{t}_i^c)$ is the orientation transformation calculated based on those key atoms of the corresponding frame extracted from $\widehat{\boldsymbol{C}}_i^u$, and the corresponding calculation process has been illustrated in Figure 3.

Then we employ the generated rotation transformation $\widehat{\boldsymbol{R}}_i$ to the corresponding $\widehat{\boldsymbol{C}}_i^{u,\text{std}}$, i.e.,

$$\widehat{\boldsymbol{C}}_i^{u,\text{rot}} = \widehat{\boldsymbol{C}}_i^{u,\text{std}} \cdot \widehat{\boldsymbol{R}}_i. \qquad (12)$$

Furthermore, the corresponding translation transformations of $\widehat{\boldsymbol{C}}_i^{u,\text{rot}}$ can be obtained through aligning the 3D coordinates of those overlapping atoms, i.e.,

$$\hat{\boldsymbol{t}}_i = \begin{cases} \boldsymbol{0}, & \text{if } i = 1, \\ \sum_{j=1}^{i-1} (\hat{\boldsymbol{c}}_{j,3}^{u,\text{rot}} - \hat{\boldsymbol{c}}_{j+1,1}^{u,\text{rot}}), & \text{if } i > 1. \end{cases} \quad (13)$$

where $\hat{\boldsymbol{t}}_i \in \mathbb{R}^3$ represents the corresponding translation transformation of $\widehat{\boldsymbol{C}}_i^{u,\text{rot}}$, $\hat{\boldsymbol{c}}_{j,3}^{u,\text{rot}} \in \mathbb{R}^3$ represents the 3D coordinate of atom-3 in $\widehat{\boldsymbol{C}}_j^{u,\text{rot}}$, and $\hat{\boldsymbol{c}}_{j+1,1}^{u,\text{rot}} \in \mathbb{R}^3$ represents the 3D coordinate of atom-1 in $\widehat{\boldsymbol{C}}_{j+1}^{u,\text{rot}}$.

Finally, we can obtain the complete polymer conformation $\widehat{\boldsymbol{C}} \in \mathbb{R}^{N \times 3}$ by employing the corresponding translation transformation $\hat{\boldsymbol{t}}_i$ to $\widehat{\boldsymbol{C}}_i^{u,\text{rot}}$, i.e.,

$$\begin{aligned} \widehat{\boldsymbol{C}}_i^{u,\text{final}} &= \widehat{\boldsymbol{C}}_i^{u,\text{rot}} + \hat{\boldsymbol{t}}_i, \\ \widehat{\boldsymbol{C}} &= \{\widehat{\boldsymbol{C}}_i^{u,\text{final}} \setminus \{\hat{\boldsymbol{c}}_{i,1}^{u,\text{final}}, \hat{\boldsymbol{c}}_{i,4}^{u,\text{final}}\}\}_{i=1}^{N_u}, \end{aligned} \quad (14)$$

where $\widehat{\boldsymbol{C}} \in \mathbb{R}^{N \times 3}$ is the complete polymer conformation, $\widehat{\boldsymbol{C}}_i^{u,\text{final}} \in \mathbb{R}^{(\frac{N}{N_u}+2) \times 3}$ is the transformed repeating unit conformation obtained by employing the corresponding translation transformation $\hat{\boldsymbol{t}}_i$, and $\setminus$ is the set difference operation.

## 4. Proposed Benchmark

In this work, we have devoted considerable time and resources to developing PolyBench, the first benchmark for polymer conformation generation. Specifically, PolyBench includes a high-quality polymer conformation dataset derived from molecular dynamics simulations (Afzal et al., 2020) and offers standardized evaluation protocols for various methods, thus setting a foundation for progress in this important yet largely unexplored research area.

### 4.1. Dataset

Since the scarcity of polymer conformation datasets is a major factor causing this important research area to remain largely unexplored, we have invested significant effort to construct a high-quality dataset of over 50,000 polymers with conformations (about 2,000 atoms per conformation) through molecular dynamics simulations (Afzal et al., 2020). In particular, the initial polymer structures of molecular dynamics simulations are generated using RDKit (Landrum et al., 2013) and AmberTools (Salomon-Ferrer et al., 2013), followed by energy minimization and equilibration in the NVT ensemble with a 1 fs time step for a total duration of 5 ns (5,000,000 steps). Besides, each simulation trajectory is obtained using the General AMBER Force Field with the GROMACS package (Van Der Spoel et al., 2005). In addition, the number of repeating units within the polymer conformation ranges from approximately 20 to 100 for most polymers, with a small portion extending beyond 100. More details about this dataset can be found in Appendix A.

### 4.2. Baseline Methods

As mentioned in Section 1 and Section 2.2, polymers significantly differ from proteins, thus rendering various protein-specific conformation generation methods unsuitable. In this context, we adapt various molecular conformation generation methods, including GeoDiff (Xu et al., 2022), TorsionalDiff (Jing et al., 2022), MCF (Wang et al., 2024b), and ET-Flow (Hassan et al., 2024), to the polymer domain as our baseline methods. Specifically, these baseline methods are implemented using their default settings, treating

*Table 1.* The performance comparison of various methods on the PolyBench benchmark

| Method | Structure | | | | Energy | | | |
| --- | --- | --- | --- | --- | --- | --- | --- | --- |
| | S-MAT-R ↓ | | S-MAT-P ↓ | | E-MAT-R ↓ | | E-MAT-P ↓ | |
| | Mean | Median | Mean | Median | Mean | Median | Mean | Median |
| GeoDiff (Xu et al., 2022) | 93.119 | 89.767 | 95.259 | 91.869 | 21.249 | 18.106 | 64.871 | 58.711 |
| TorsionalDiff (Jing et al., 2022) | 53.210 | 38.710 | 70.679 | 60.744 | 2.605 | 1.034 | 8.402 | 6.851 |
| MCF (Wang et al., 2024b) | 248.432 | 242.866 | 258.891 | 253.239 | | $> 10^{10}$ | | |
| ET-Flow (Hassan et al., 2024) | 94.057 | 90.475 | 96.896 | 92.877 | 6.733 | 5.186 | 53.528 | 30.125 |
| PolyConf (ours) | **35.021** | **24.279** | **46.861** | **37.996** | **0.933** | **0.359** | **6.191** | **4.122** |

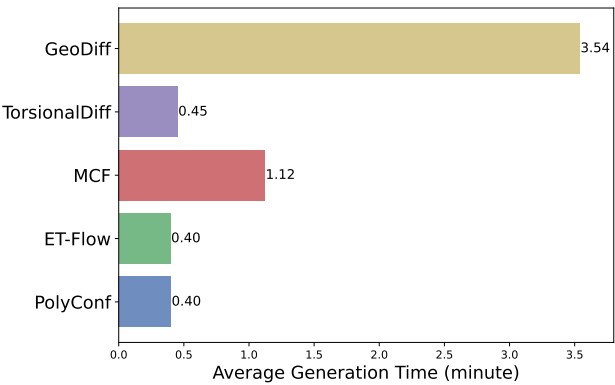

*Figure 5.* The efficiency comparison (average time) of various methods on the PolyBench benchmark, where we compare the average time of generating polymer conformations.

polymers as large molecules containing more atoms. Please note that TorsionalDiff requires an initial polymer structure as input, which cannot be directly generated like small molecules using RDKit (Landrum et al., 2013), we have to employ the initial polymer structure of the corresponding simulation trajectory as its input, thus unintentionally giving TorsionalDiff a biased advantage over other methods.

### 4.3. Evaluation Metrics

To guarantee the comprehensive and fair comparison, we evaluate various methods based on both *structure* and *energy* perspectives. Here, $S_g$ and $S_r$ denote the sets of generated and reference conformers, respectively. The *structure* metrics are defined as follows:

$$\text{S-MAT-R} = \frac{1}{|S_r|} \sum\nolimits_{C \in S_r} \min_{\widehat{C} \in S_g} \text{RMSD}(C, \widehat{C}),$$
$$\text{S-MAT-P} = \frac{1}{|S_g|} \sum\nolimits_{\widehat{C} \in S_g} \min_{C \in S_r} \text{RMSD}(C, \widehat{C}),$$

(15)

where the generated conformer $\widehat{C}$ and reference conformer $C$ have already been aligned before computing their RMSD.

Meanwhile, the *energy* metrics are defined similarly by replacing the structural difference $\text{RMSD}(C, \widehat{C})$ in Eq. (15) with potential energy difference $|E(C) - E(\widehat{C})|$.

**Remark.** We think the widely used **Coverage** metric (Xu et al., 2022), relying on a fixed RMSD threshold $\delta$ for structural comparison in the small molecule domain, is unsuitable for polymer conformation generation as polymers typically exhibit a much larger conformational space with significant diversity arising from their chain length, flexibility, and repeating units (Chen et al., 2021). Thus, we exclude it from our evaluation metrics but still report the corresponding performance under this metric in Appendix B for reference.

### 4.4. Main Results

Table 1 summarizes the performance of various methods on the PolyBench benchmark, demonstrating that our Poly-Conf consistently outperforms baseline methods across both structure (S-MAT-R, S-MAT-P) and energy (E-MAT-R, E-MAT-P) metrics with a significant margin. In particular, compared to TorsionalDiff (i.e., the best baseline), our Poly-Conf achieves substantial improvements with at least 25% across all evaluation metrics, while **eliminating the need for a pre-determined polymer structure**, highlighting its effectiveness and practicability for generating polymer conformations that are both structurally accurate and energetically realistic. Besides, the suboptimal performance of baseline methods also underscores their limitations in capturing the structural complexity of polymers. The above results further validate and demonstrate the critical need for designing tailored polymer conformation generation methods, such as our PolyConf, to accommodate the unique conformational intricacies of polymer systems.

Meanwhile, we compare the efficiency of various methods on the PolyBench benchmark, as illustrated in Figure 5. In particular, the average time required to generate polymer conformations across various methods is computed and compared. Our PolyConf achieves the fastest generation time of 0.40 minutes, surpassing TorsionalDiff (0.45 minutes), MCF (1.12 minutes), and GeoDiff, which is significantly slower at 3.54 minutes. Combined with Table 1, these results demonstrate that our PolyConf is capable of both effective and efficient polymer conformation generation.

In addition, we further present some visualization examples of polymer conformations generated by our PolyConf and

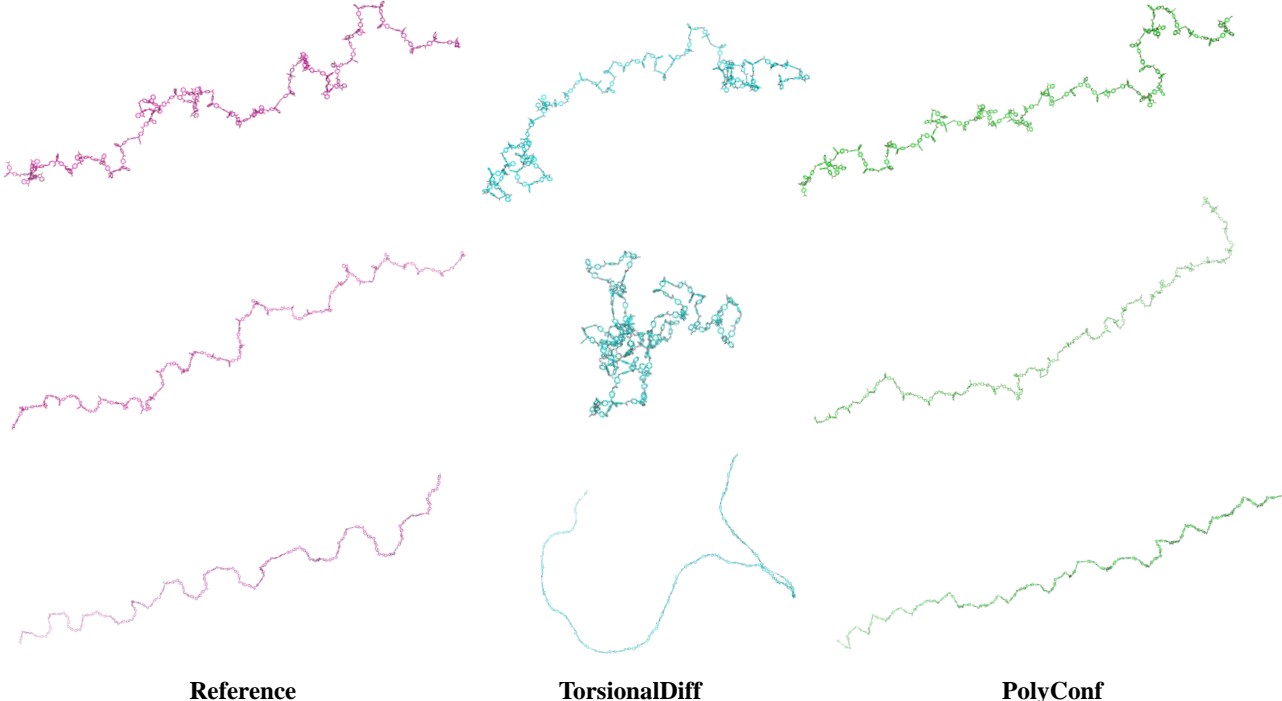

**Reference**          **TorsionalDiff**          **PolyConf**

*Figure 6.* Several visualization examples of our PolyConf and TorsionalDiff (i.e., the best baseline).

*Table 2.* The scalability evaluation of various methods, where the number of repeating units per polymer is doubled.

| Method | Structure | | | | Energy | | | |
| --- | --- | --- | --- | --- | --- | --- | --- | --- |
| | S-MAT-R ↓ | | S-MAT-P ↓ | | E-MAT-R ↓ | | E-MAT-P ↓ | |
| | Mean | Median | Mean | Median | Mean | Median | Mean | Median |
| GeoDiff (Xu et al., 2022) | 184.668 | 175.607 | 186.861 | 177.645 | 52.614 | 47.872 | 112.883 | 105.197 |
| TorsionalDiff (Jing et al., 2022) | 119.289 | 94.075 | 146.816 | 126.932 | 5.219 | 2.216 | 11.692 | 9.227 |
| MCF (Wang et al., 2024b) | 227.691 | 252.796 | 280.805 | 260.882 | \multicolumn{4}{c}{$> 10^{10}$} | | | |
| ET-Flow (Hassan et al., 2024) | 186.132 | 176.370 | 188.725 | 178.977 | 15.331 | 12.465 | 65.116 | 41.642 |
| PolyConf (ours) | **65.040** | **41.992** | **84.626** | **64.445** | **1.259** | **0.609** | **5.785** | **4.434** |

the best baseline in Figure 6 to provide qualitative insights. It demonstrates that our PolyConf produces polymer conformations that more closely align with references, confirming its capability to generate high-quality polymer conformations. In contrast, despite leveraging biased prior knowledge of the initial polymer structure, TorsionalDiff (i.e., the best baseline) still fails to capture the unfolded and relaxed polymer conformations effectively.

### 4.5. Scalability Evaluation

As introduced in Section 1, polymers are macromolecules formed by the covalent bonding of numerous identical or similar monomers. Due to the nature of polymerization, a single polymer chain can vary in length depending on specific reaction conditions, resulting in a chain length distribution rather than a uniform chain length. Consequently, the same polymer can exhibit conformations at different scales,

determined by the number of repeating units incorporated during polymerization. Given this inherent variability, it's also essential to evaluate the scalability of various methods, particularly their ability to generate polymer conformations with larger scales (i.e., more atoms or repeating units).

Considering various models in Table 1 are trained on polymer conformations comprising approximately 2,000 atoms, we further apply these trained models to generate conformations with roughly 4,000 atoms by simply doubling the number of repeating units in the inference phase. Here, we use the same pipeline of molecular dynamics simulations to generate enlarged polymer conformations on the test set as references. The corresponding evaluations are performed using both structural metrics (S-MAT-R, S-MAT-P) and energy metrics (E-MAT-R, E-MAT-P), as summarized in Table 2. Owing to the advantages of masked autoregressive modeling, our PolyConf demonstrates excellent performance,

significantly outperforming baseline methods across all evaluation metrics. In particular, compared to TorsionalDiff (i.e., the best baseline), our PolyConf improves energy metrics by over 50%. These results validate and support the superior scalability and generalization capabilities of our PolyConf, thus setting it apart as an effective method for varying-scale polymer conformation generation.

## 5. Conclusion

In this work, we successfully unlock a critical yet largely unexplored task in the context of machine learning — polymer conformation generation, which may further trigger a series of downstream research directions. Specifically, we propose PolyConf, the first tailored polymer conformation generation method that leverages hierarchical generative models to tackle this task, and develop PolyBench, the first benchmark for polymer conformation generation, to overcome the scarcity of polymer conformation datasets and boost subsequent studies. Extensive and comprehensive experiments on the PolyBench benchmark consistently demonstrate that our PolyConf significantly outperforms existing conformation generation methods in both quality and efficiency while maintaining superior scalability and generalization capabilities, thus highlighting its exceptional ability in polymer conformation generation.

In the future, we will continue to explore this important research area and further develop our method and benchmark, which may inspire broader research efforts and drive progress in this research area. Moving forward, we aim to extend our method to accommodate more intricate polymer architectures, such as those with 2D topological structures and those constructed by more than one kind of monomer. Meanwhile, we will also consider integrating other advanced deep generative models (e.g., flow-based generative models) to refine polymer conformation generation.

## Acknowledgements

This work was supported by the National Natural Science Foundation of China (92270110), the Fundamental Research Funds for the Central Universities, the Research Funds of Renmin University of China, and the Public Computing Cloud, Renmin University of China. We also acknowledge the support provided by the fund for building world-class universities (disciplines) of Renmin University of China and by the funds from Beijing Key Laboratory of Research on Large Models and Intelligent Governance, Engineering Research Center of Next-Generation Intelligent Search and Recommendation, Ministry of Education, and from Intelligent Social Governance Interdisciplinary Platform, Major Innovation & Planning Interdisciplinary Platform for the "Double-First Class" Initiative, Renmin University of China.

## Impact Statement

This paper presents work that aims to advance the field of Machine Learning and AI for Science, especially for polymer conformation generation. There are many potential societal consequences of our work, none of which we feel must be specifically highlighted here.

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

# A. Dataset Details

In this work, considering the scarcity of polymer conformation datasets is a major factor causing this important research area to remain largely unexplored, we have dedicated significant time and resources to developing a high-quality dataset comprising over 50,000 polymers with their conformations (about 2,000 atoms per conformation) obtained through molecular dynamics simulations (Afzal et al., 2020). Here, the specific pipeline of molecular dynamics simulations is described in Appendix A.1, while Appendix A.2 provides detailed statistics of the obtained polymer conformation dataset.

## A.1. Construction Pipeline

While previous studies have introduced DFT-based conformation datasets for polymers (Feng et al., 2023), these datasets are limited to systems with no more than six repeating units, which is far from the realistic scenario where polymer systems can consist of thousands of atoms. Besides, considering the prohibitively high computational cost of DFT calculations for polymer systems of this size (about 2,000 atoms in our work), we employ force-field-based simulations for polymer conformation generation. Although force fields may have some limitations in accurately predicting properties or energetics, they are well-suited for efficiently exploring the conformational space of large polymer systems, thereby ensuring the generated conformations are realistic representations of polymer structures without compromising the focus of our task.

In particular, polymer systems are constructed and prepared for molecular dynamics (MD) simulations through combining various molecular modeling tools and custom Python scripts. Monomer structures are generated by RDKit (Landrum et al., 2013) based on the corresponding SMILES strings and then processed to define chain termini and repeat units. The polymer consists of a repeating PET unit, capped by head (HPT) and tail (TPT) units. RDKit is utilized to analyze atom connectivity and identify key atoms for polymerization. Hydrogen atoms at the connection points are omitted, and chain termini are explicitly parameterized. Atomic charges and topology files for the corresponding monomer units are generated using the Antechamber and prepgen tools (Salomon-Ferrer et al., 2013), employing the General AMBER Force Field (GAFF). TLeap is used to create AMBER-compatible topology and coordinate files for both individual monomers and polymer chains. Polymer chains with a degree of polymerization ($N_u$) are constructed by repeating the PET unit $N_u - 2$ times and capping the chain with HPT and TPT units at the termini. Chain lengths are chosen to achieve approximately 2,000 atoms per system. AMBER topology and coordinate files are then converted to GROMACS-compatible formats using ACPYPE. The resulting files are then organized for subsequent MD simulations.

The optimization and MD simulations are conducted using GROMACS (Van Der Spoel et al., 2005) as the usage of GROMACS on polymer simulations has long been reported (Liu et al., 2024; Grünewald et al., 2022). Specifically, the steepest descent method is applied to minimize the system energy, and then 5,000,000 steps (5ns) of MD calculations are performed at 298 K and 1 atm using the NVT ensemble for equilibrium calculations.

## A.2. Dataset Statistics

We collect 1D polymer SMILES strings from various publicly available sources, and further divide their corresponding 3D conformations obtained through the molecular dynamics simulations into training, validation, and test sets. Specifically, the training set comprises about 46k polymers with their corresponding conformations, the validation set comprises about 5k polymers with their corresponding conformations, and the test set comprises about 2k polymers with their corresponding conformations. Meanwhile, as illustrated in Figure 3, the polymer conformation can be decomposed into a series of repeating unit conformation, so we visualize the distribution of the number of repeating units per polymer conformation in Figure 7 to provide insights into the structural complexity and variability of the polymer conformations. As shown in this figure, the number of repeating units ranges from approximately 20 to 100 for most polymers, with a small portion extending beyond 100. The distribution across the training, validation, and test sets demonstrates the ability of our dataset to capture a wide variety of polymer structures, ensuring diverse and representative conformations for robust evaluation.

# B. Additional Results

In the molecular conformation generation task (Xu et al., 2022), Coverage (COV) and Matching (MAT) metrics are widely used to evaluate the diversity and quality of generated conformations. Through a fixed RMSD threshold, the Coverage metric can measure the fraction of reference conformations captured by the generated conformations (recall perspective) or the fraction of generated conformations captured by reference conformations (precision perspective). However, its applicability to polymer conformation generation is limited due to the inherent flexibility, polydispersity, and amorphous

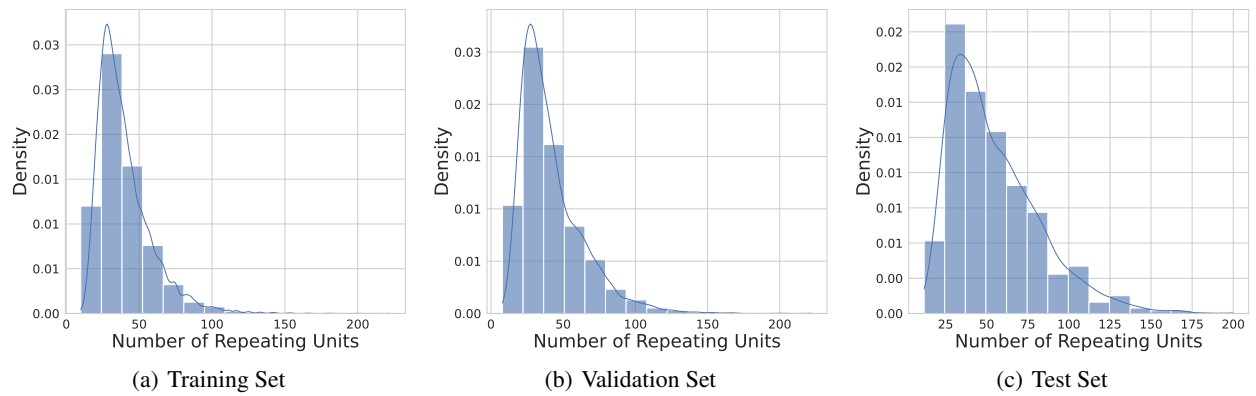

|           (a) Training Set           |          (b) Validation Set          |            (c) Test Set             |

*Figure 7.* The distribution of the number of repeating units per polymer conformation.

*Table 3.* The structural quality of generated polymer conformations in terms of Coverage (%) and Matching (Å). We compute Coverage with a threshold of $\delta = 25$ Å to better distinguish top methods.

| Method | Recall | | | | Precision | | | |
|--------|--------|--------|--------|--------|--------|--------|--------|--------|
|        | S-COV-R ↑ | | S-MAT-R ↓ | | S-COV-P ↑ | | S-MAT-P ↓ | |
|        | Mean | Median | Mean | Median | Mean | Median | Mean | Median |
| GeoDiff (Xu et al., 2022) | 0.108 | 0.000 | 93.119 | 89.767 | 0.008 | 0.000 | 95.259 | 91.869 |
| TorsionalDiff (Jing et al., 2022) | 0.172 | 0.000 | 53.210 | 38.710 | 0.100 | 0.000 | 70.679 | 60.744 |
| MCF (Wang et al., 2024b) | 0.000 | 0.000 | 248.432 | 242.866 | 0.000 | 0.000 | 258.891 | 253.239 |
| ET-Flow (Hassan et al., 2024) | 0.089 | 0.000 | 94.057 | 90.475 | 0.064 | 0.000 | 96.896 | 92.877 |
| PolyConf (ours) | **0.515** | **1.000** | **35.021** | **24.279** | **0.336** | **0.100** | **46.861** | **37.996** |

nature of polymer systems. Therefore, we exclude the Coverage metric from our evaluation metrics in Section. 4.3, but still report the corresponding performance of various methods under this metric for reference here.

As shown in Table 3, our PolyConf still significantly outperforms baseline methods in both structural Coverage (S-COV) and Matching (S-MAT) metrics. In particular, our PolyConf achieves the highest S-COV-R (Recall) of 0.515 (mean) and 1.000 (median), demonstrating superior diversity and structural coverage over the reference set. Additionally, it achieves the lowest S-MAT-R of 35.021 Å(mean) and 24.279 Å(median), indicating a closer match to the reference conformations compared to baseline methods, which show much higher deviations. In terms of precision-based metrics (S-COV-P and S-MAT-P), our PolyConf also maintains the strong superiority with 0.336 S-COV-P (mean) and 0.100 (median), while achieving the best S-MAT-P of 46.861 Å(mean) and 37.996 Å(median). These results consistently indicate that our PolyConf generates polymer conformation with both broader coverage and more accurate conformational matching, despite the inherent difficulties posed by the flexibility and variability of polymer systems.

