# OpenReview forum: "PolyConf: Unlocking Polymer Conformation Generation through Hierarchical Generative Models"
_ICML.cc/2025/Conference — ICML 2025 poster_

### Official Review · Reviewer_QQYG · 2025-02-27

**Overall Recommendation:** 3

**Summary:**

In this work, we proposed PolyConf, a pioneering tailored polymer conformation generation method that leverages hierarchical generative models to unlock new possibilities for this task. The authors decompose the polymer conformation into a series of local conformations generating these local conformations through an autoregressive model. The Polymers contain multiple repeats, PolyConf first generates one unit and then generates a series of transformations to locate each repeat. Experimental results demonstrate that PolyConf can generate high-quality, physically reliable polymer conformations, facilitating advancements in polymer modeling and simulation.

## update after rebuttal
The authors successfully addressed my concerns. I suggested the authors should clearly pointed out this model is designed specific to the material science. The overall contribution from the machine learning aspect is limited but there are couple of technical improvements for this new task. Therefore, I decide to maintain my overall ranking as 'weak accept'.

**Claims And Evidence:**

Yes

**Essential References Not Discussed:**

No.

**Experimental Designs Or Analyses:**

Yes. I check all the results in the main article and appendix

**Methods And Evaluation Criteria:**

Yes

**Other Comments Or Suggestions:**

Related to the previous question, does the polymer have a stable 3D structure? Is there any biochemistry or application related to it? Could you provide some specific citations for its applications?

**Other Strengths And Weaknesses:**

First, I would like to thank the authors for contributing a new dataset of polymer 3D structures to the field. In terms of the generated 3D structures, PolyConf clearly has significant advantages over other methods. However, in terms of model construction, there isn’t much innovation in this work. The authors split the generation of a repeat polymer into two parts: the first step is to generate a small molecule’s 3D structure, which has been extensively studied, and the authors should compare different methods for predicting small molecule 3D structures. The second step involves generating the repeat units, which, as far as I know, is new. Therefore, from a purely modeling perspective, the contribution of this work is limited.

Additionally, during the generation of repeats, each repeat is created independently, at least according to equations (9) and (10). Why would this modeling approach, which seems to lack global information, lead to a reduction in overall RMSD? My guess is that if a repeat is stretched into a straight line, the RMSD won't be too high. Therefore, the second step of the model generation does not necessarily need to learn any special patterns. The authors could provide different 3D structures to address this concern.

**Questions For Authors:**

1.What is the meaning of ‘To generate a random subset of unknown repeating unit conformations based on known/predicted repeating unit conformations iteratively (expressed in Eq. (3)) ‘  what is the exact meaning of ‘unknown’. Does it mean we do not know the 3D structure or we do not know the 2D structure?

2.Is each R-i independent? If they are all independent, could there be collisions between different repeats when the number of repeats is large?

3. What is the RMSD between different conformations in the MD simulation? One possibility is that these polymers themselves do not have a stable 3D structure, and there is a lot of structural variation in the MD simulations. For example, if the RMSD among the MD simulated structures is ~100, then an RMSD of 30 obtained from one computational method would be meaningless.

**Relation To Broader Scientific Literature:**

This part I am not quite sure. I am asking the authors to provide such evidence. I am not quite aware any biological and chemical applications that the prediction of 3D structure of a polymer is necessary.

**Theoretical Claims:**

Yes

---

> ### Author Rebuttal · Authors · 2025-03-31
>
> Thanks for your comments. Below, we try to resolve your concerns one by one.
>
> **W1: Clarification of model construction and contribution**
>
> As shown in Figure 2, we design PolyConf as a hierarchical generative framework with a two-phase generating process.
>
> As described in Section 3.2, in the first phase, we leverage the masked autoregressive model (MAR) to generate the conformation of each repeating unit within the given polymer in random order. **Please note that the conformation of each repeating unit within the given polymer is not generated independently. They share the same global 2D polymer graph information and can be influenced by each other through the MAR module.** The code is provided in `./polyconf/models/polyconf_phase1.py` within the anonymous repository (https://anonymous.4open.science/r/PolyConf).
>
> As described in Section 3.3, in the second phase, we employ an SO(3) diffusion model to generate the corresponding orientation transformations of repeating units within the given polymer, thereby assembling those repeating unit conformations generated by the previous phase into the complete polymer conformation. **Please note that the corresponding orientation transformations of repeating units within the given polymer are obtained together through the diffusion processes on $SO(3)^{Nu}$.** The code is provided in `./polyconf/models/polyconf_phase2.py` within the anonymous repository (https://anonymous.4open.science/r/PolyConf).
>
> Here, as mentioned in lines 79-100, PolyConf is specifically designed based on the unique characteristics of polymers, not a simple application of the existing methods. Besides, as shown in our response to Reviewer HPGV (https://openreview.net/forum?id=BsTLUx38qV&noteId=AdOXTd7M9Y), PolyConf can achieve even better performance than the SOTA polymer property prediction method, further demonstrating its potential. In addition, we think PolyConf also has significant potential for other macromolecules composed of building blocks, such as proteins (amino acids) and RNA (nucleotides), thereby driving progress in these related fields.
>
>
> **W2&Q2: Clarification of modeling independence**
>
> As mentioned in our last response, both conformation and orientation transformation of each repeating unit within the given polymer **are not generated independently**. They share the same global information provided by the 2D polymer graph and can be influenced by each other.
>
>
> **W3&Q3: Stable 3D structures and their applications**
>
> We analyzed the energy changes of polymers during our MD simulations, and the results show that most simulations achieve convergence within 1 ns, proving that polymer conformations obtained through our MD simulations are low-energy states (i.e., stable).
> As shown in the following Table, we further calculate the RMSD between conformation obtained in 2ns/3ns/4ns and the final conformation obtained in 5ns within the same MD tracjory, demonstrating that the polymer has a stable 3D structure.
> | RMSD  | 2ns       | 3ns       | 4ns       |
> |---|---|---|---|
> | 5ns   | 2.15 ± 1.22 | 2.01 ± 1.09 | 1.84 ± 0.97 |
>
> These stable 3D structures are critical for various applications. For example, the work in [1] has revealed the relation between the polymer conformation and the elastic modulus of the crystalline region. The works in [2][3] have tried to apply them to calculate glass transition temperatures through experiments and MD simulations. The work in [4] has incorporated polymer 3D structural information into property prediction. In addition, as shown in our response to Reviewer HPGV (https://openreview.net/forum?id=BsTLUx38qV&noteId=AdOXTd7M9Y), PolyConf can achieve even better performance than the SOTA polymer property prediction method, further demonstrating the importance and potential of polymer conformation.
>
>
> **Q1: Clarification of Eq.3**
>
> As mentioned in our first response, we leverage the masked autoregressive model to generate the conformation of each repeating unit within the given polymer in random order during the first phase. Here, the 2D structure information is known, and we aim to generate the 3D structures of the unpredicted repeating units based on the predicted repeating unit iteratively, thereby obtaining the 3D structures of all repeating units within the given polymer.
>
> We hope the above responses can resolve your concerns. Since it's a relatively unexplored research area, we kindly request your understanding of the challenges and complexities involved in our work.
>
> [1] Relation between the polymer conformation and the elastic modulus of the crystalline region of polymer. Journal of Polymer Science Part C, 1970.
>
> [2] Prediction of polymer properties. cRc Press, 2002.
>
> [3] High-throughput molecular dynamics simulations and validation of thermophysical properties of polymers for various applications. ACS Applied Polymer Materials, 2020.
>
> [4] MMPolymer: A multimodal multitask pretraining framework for polymer property prediction. CIKM2024.

---

> > ### Comment · Reviewer_QQYG · 2025-04-03
> >
> > Thanks for clarifying the technical details. I now understand the entire generation process. The generation of repeat units is new.
> > On the other hand, you mentioned further applications such as proteins, RNAs and DNAs. There are already tons of algorithms which generate 3D structures (w/o sequences). To motivate this work, it is better to show up real biological applications. Therefore, I would like to main my score as weak accept.

---

> > > ### Author Response · Authors · 2025-04-04
> > >
> > > Thanks for your timely feedback. **Here, we want to further clarify the main contributions of this work**.
> > >
> > > As we have discussed in the whole paper, **our work focuses on polymers in material science rather than proteins in biology**, aiming to explore polymer conformation generation, **an important yet unexplored research area in material science**. As shown in lines 80-98, compared with proteins and other bio-complex, polymers have their unique challenges, e.g., greater structural flexibility. For this goal, we have devoted considerable time and resources to developing PolyBench, **the first benchmark** for polymer conformation generation, to address the scarcity of polymer conformation datasets. Furthermore, we propose PolyConf, **the first tailored method** for polymer conformation generation, which can consistently generate high-quality, physically reliable polymer conformations, facilitating advancements in polymer modeling and simulation. Besides, **the whole work, including code, model, and data, will all be publicly available** to boost subsequent studies. In addition, as shown in our response to Reviewer HPGV (https://openreview.net/forum?id=BsTLUx38qV&noteId=AdOXTd7M9Y), our work can achieve even **better polymer property prediction performance** than the SOTA method, further demonstrating its potential in material modeling and design.
> > >
> > > Although we mentioned in the rebuttal that our method has the potential for modeling conformations of proteins, RNAs, and DNAs, these applications are not our core contributions. **Exploring these applications can be our future work, but they are out of the scope of this work.**
> > >
> > >
> > > According to ICML 2025 Reviewer Instructions, the discussion between authors and reviewers is restricted to at most one additional round of back-and-forth, which means that we might no longer have the opportunity to respond to any further feedback from you. **We hope the above response helps to further clarify the focus and main contributions of our work and enhances your confidence to further support our work. We would greatly appreciate it if you could consider raising your score based on our contributions.** Thanks in advance for your consideration.
> > >
> > >
> > > Regards,
> > >
> > > The authors of PolyConf

---

### Official Review · Reviewer_HPGV · 2025-03-11

**Overall Recommendation:** 3

**Summary:**

Deep learning for polymer design is a severely underexplored area, and this work attempts to address two major challenges: the lack of methodology and the scarcity of high-quality data. In PolyConf, the authors generate polymers autoregressively, generating conformers for individual building blocks and linking them using a diffusion model. This study introduces the first benchmark with a (supposedly) high-quality polymer conformation dataset derived from MD simulations, aiming to advance research in this area. The work is novel and if reproducible would be highly impactful, however, I have quite a few questions and concerns, which I outline below.

The work aims to address the lack of polymer conformer data by publishing a new dataset; however, I have been unable to review the dataset as it was not made available with the paper (even anonymously), making it difficult to validate its quality. Another limitation of the current evaluation is that all methods compared in the paper are designed for small molecules rather than polymers; however, as far as I understand, there are no deep generative models available for polymers that also handle 3D information, so this may be acceptable. While the contributions are promising and the figures are well-designed and informative, I have currently rated this paper marginally below the publication threshold. I would be willing to increase my score if the authors provide anonymized code for review and address the points raised below.

Feedback and Questions

- **Code availability:** No code is provided, which is a significant drawback. Releasing code (even anonymously for review) would greatly improve reproducibility and impact, especially considering that one of the contributions if the new dataset, but I cannot evaluate it if it is not presented for review. Lack of reproducibility is one of the main reasons I have ranked this work below the acceptance threshold, despite its significant novelty. Without code, it is difficult to determine if the work is fully reproducible, and I am not sure there are enough details in the text to accurately reproduce the results.

- **Baseline comparisons:** Are there any non-deep-learning baselines for comparison? For instance, a simpler approach that links building block conformers could serve as a useful reference, more-so than comparing to deep generative models designed for small molecules. What would the RMSD be for a method that simply linked the conformers generated by RDKit for the same SMILES in the dataset? This could serve as an interesting dummy model for comparison.

- **Metric clarity:**
    - The units for the S-MAT-R and S-MAT-P metrics (both structural and energy-related) should be explicitly stated, I have no idea currently what the units are for the tables in the paper which makes it hard to assess how bad/good the values are.
    - It is also unclear if the metrics reported in the paper (in the various tables) are for the training, validation, or test set, nor how the metrics differed for the different sets. This should be clarified and unambigious. Have the authors demonstrated, furthermore, that their model is not overfitting?

- **Dataset details:**

    - The authors mention sourcing data from three different sources. How much data came from each?

    - PolyInfo is known to prohibit web scraping, so I am confused as to how was data obtained from this source?

    - A dimensionality reduction method (e.g., UMAP) could help visualize the overlap in building block SMILES from each source, and would make for an interesting analysis to tell us how different the building blocks from each source are.

- **Validation of MD simulations:** How were the MD simulations used for dataset construction validated? It would be useful to understand how reliable they are for training deep learning models.

- **Masked autoencoder experiments:** Was there any analysis on how different levels of masking (e.g., percent of masked bits) affect performance? Understanding the limits of masking would be valuable, especially since I assume that increased masking in the latent space would improve inference efficiency. If this is not the case, some clarification would be helpful.

- **RMSD calculations:**

    - Were generated and reference polymer structures aligned before computing RMSD?

    - If so, the reported RMSD values appear high. However, without units provided, it is difficult to assess this properly.

- **Failure mode analysis:** Can the authors highlight any failure cases? For instance, can the model currently only handle linear polymers, or does it also support other topologies? Are there specific polymers or building blocks where the model struggles?

- **Objective clarification:** Is the goal to generate low-energy conformers or to match reference structures? If the latter, have the authors verified that the reference conformers are indeed low-energy states?

- **Future directions:** Could the learned embedding from the masked autoencoder-decoder framework be useful for polymer property prediction? This might be an interesting avenue for future work.

**Claims And Evidence:**

I do not believe the claims are justified by the current presented results. See my detailed review above.

**Essential References Not Discussed:**

N/A

**Experimental Designs Or Analyses:**

The analyses in this work are partially sound but require significant clarification. See my detailed review above.

**Methods And Evaluation Criteria:**

Partially. See my detailed review above.

**Other Comments Or Suggestions:**

See my detailed review above.

**Other Strengths And Weaknesses:**

Significance is high and the method novel and creative, but the clarity is weak and the experiments/analysis lack rigour. This can be potentially improved.

**Questions For Authors:**

See my detailed review above.

**Relation To Broader Scientific Literature:**

This is a novel work that addresses a key gap in polymer design via deep generative models.

**Theoretical Claims:**

N/A

---

> ### Author Rebuttal · Authors · 2025-03-31
>
> Thanks for your comments and suggestions. Below are our responses to your questions.
>
> **Q1&Q5: Code availability and Validation of MD**
>
> The code, data, and various scripts for our PolyConf and MD simulations are available in this anonymous link (https://anonymous.4open.science/r/PolyConf). It has provided enough details to reproduce our work. Due to the repository size limitation, we can only provide a small subset of the complete dataset, but the complete dataset will be provided after acceptance.
>
> As mentioned in our response to Reviewer 9jKE (https://openreview.net/forum?id=BsTLUx38qV&noteId=KDYQcy2SLb), we have invested significant effort to validate our MD simulations, including seeking guidance from experienced experts, examining the energy convergence, calculating the density values of typical polymers, and comparing them with experiment density values.
>
>
> **Q2: Baseline comparisons**
>
> As shown in the following Table, we have constructed such a dummy model following your suggestion, and the results demonstrate that our PolyConf can still achieve SOTA performance.
> | Model         | S-MAT-R   | E-MAT-R   |
> |----|---|---|
> | Dummy Model   | 68.403    | 18.735    |
> | PolyConf      | **35.021**    | **0.933**     |
>
>
> **Q3: Metric clarity**
>
> The units for the S-MAT-R and S-MAT-P are Å, and the metrics reported are all for the test set. We will explicitly state them in the revised paper.
>
> In addition, we train PolyConf on the training set and select the best checkpoint based on the validation set. It is a widely used practice to avoid overfitting.
>
>
> **Q4: Dataset details**
>
> The details can be found in Appendix A, with the majority derived from PI1M [1]. Please note that we only need to collect polymer SMILES strings, and all these strings are publicly available from previous works [1]-[5].
>
>
> **Q6: Masked autoencoder experiments**
>
> In our implementation, the mask rate is randomly sampled from a pre-defined truncated normal distribution, ensuring balanced randomness, avoiding extreme values, and enhancing both robustness and generalization. Details can be found in `./polyconf/models/polyconf_phase1.py` within the provided anonymous repository.
>
>
> **Q7: RMSD**
>
> The generated and reference polymer structures have been aligned before computing RMSD. While PolyConf has achieved SOTA performance, there is still significant room for improvement in terms of the RMSD. In the future, we are willing to collaborate with researchers to further explore this challenging task.
>
>
> **Q8: Failure mode analysis**
>
> Our method currently only handles linear polymers, as modeling other topologies (e.g., cross-linked polymers) involves significantly greater complexities beyond data modeling.
>
>
> **Q9: Objective clarification**
>
> We have analyzed the energy changes of polymers during our MD simulations, and the results show that most simulations achieve convergence within 1 ns, while we run the simulations for 5 ns to ensure the reference conformers are low-energy states. Here, we have provided some raw outputs of our MD simulations in the `./MD` folder within the provided anonymous repository for validation.
>
> Therefore, the objectives you mentioned are fundamentally the same: we aim to train the model to match reference structures, thereby enabling it to generate low-energy conformers.
>
>
> **Q10: Future directions**
>
> As shown in the following Table, directly based on the learned embedding, PolyConf can achieve even better performance than the SOTA polymer property prediction method MMPolymer [5], further demonstrating its great potential.
> | Method      | Egc        | Egb        | Eea        | Ei         | Xc         | EPS        | Nc         | Eat        |
> |---|---|---|---|---|---|---|---|---|
> | MMPolymer   | **0.924 ± 0.006** | 0.934 ± 0.008 | 0.925 ± 0.025 | **0.836 ± 0.053** | **0.488 ± 0.072** | 0.779 ± 0.052 | 0.864 ± 0.036 | **0.961 ± 0.018** |
> | PolyConf    | 0.916 ± 0.006 | **0.937 ± 0.010** | **0.926 ± 0.018** | 0.822 ± 0.052 | 0.422 ± 0.096 | **0.811 ± 0.049** | **0.868 ± 0.041** | **0.961 ± 0.030** |
>
>
> We hope the above responses help you re-evaluate our work. Since polymer conformation generation remains a relatively unexplored research area, we have invested significant time and effort into developing and refining our PolyConf and PolyBench. We kindly request your understanding of the challenges and complexities involved in pioneering work. We would greatly appreciate it if you could raise your score.
>
> [1] PI1M: a benchmark database for polymer informatics. Journal of Chemical Information and Modeling, 2020.
>
> [2] Graph rationalization with environment-based augmentations. KDD2022.
>
> [3] Polymer informatics at scale with multitask graph neural networks. Chemistry of Materials, 2023.
>
> [4] PolyNC: a natural and chemical language model for the prediction of unified polymer properties. Chemical Science, 2024.
>
> [5] MMPolymer: A multimodal multitask pretraining framework for polymer property prediction. CIKM2024.

---

> > ### Comment · Reviewer_HPGV · 2025-04-03
> >
> > Thank you to the authors for the thoughtful response and for including a link to the anonymized repo. While I understand the size limitations, the code is missing a README and general documentation (e.g., set-up instructions) that limits its utility significantly as well as the reproducibility (if there are no instructions for how to reproduce results from the paper, then one cannot really count it as "reproducible"). I recommend including these in the revisions. Furthermore, for data from MD simulations, this does not belong in a Git repo, as it is not made to handle large files, but in a separate Zenodo (or similar platform) with also suitable accompanying documentation.
> >
> > Thank you also for the additional details, it is interesting. If the additional analysis and discussion discussed above in the author rebuttal is also incorporated into the manuscript in a structured way, and all my original questions are answered (some, especially around the data curation and visualization, were ignored) I will consider increasing my score to a 3.

---

> > > ### Author Response · Authors · 2025-04-03
> > >
> > > Thanks for your timely feedback.  Below, we try to resolve your remaining concerns one by one.
> > >
> > > > **1. For the anonymized repo:**
> > >
> > > **According to the ICML 2025 Peer Review FAQ, it's forbidden to include additional text in the code.**
> > > That's why we have to remove the README.md and general documentation from our anonymized repo.
> > >
> > > Here, we will provide a brief introduction to reproduce our work:
> > > - For environment set-up, the list of dependencies has been provided in the accompanying `requirements.txt` file.
> > > - For Phase-1 Training, please run `bash train_phase1.sh`.
> > > - For Phase-2 Training, please run `bash train_phase2.sh`.
> > > - For inference, please run `bash inference.sh`.
> > > - For generated conformations extraction, please run `python extract_confs.py`
> > >
> > > **We will release our code with detailed documentation in the public version, as we have always done before.**
> > >
> > > > **2. For MD simulation data:**
> > >
> > > Thanks for your suggestion.
> > > **Since the total size of the MD data is as large as around 5TB, it is impractical for us to release such a dataset at the current stage.** To solve your concern with our best effort under this context, we have randomly sampled 50 cases from our MD data. Please access these sample data at this link (https://drive.google.com/file/d/1kZJDR_oIJq98xa7TZuuhkAhjTD89Q1px/view?usp=drive_link), and we will release the whole MD data with detailed documentation after acceptance.
> > >
> > > In addition, **we kindly remind you that the corresponding scripts of our MD simulations have already been provided in the `./MD` folder of our anonymized repo** (https://anonymous.4open.science/r/PolyConf).
> > > After installing AmberTools and GROMACS according to the corresponding official documentation, you can easily reproduce the pipeline of our MD simulations through `python prepare_md.py` and `python run_nvt_md.py`.
> > >
> > > Therefore, we believe that the delay in releasing the whole MD data is not strong evidence for rejecting our work.
> > >
> > > > **3. For manuscript revision:**
> > >
> > > **According to the ICML 2025 Peer Review FAQ, it's also forbidden to update the original submission (PDF and supplemental material) during the discussion period.** Here, we promise that the additional analysis and discussion in the rebuttal will all be incorporated into our camera-ready version in a structured way.
> > >
> > > > **4. For data curation and visualization (Q4. Dataset details):**
> > >
> > > **4.1 The authors mention sourcing data from three different sources. How much data came from each?**
> > >
> > > As we have responded in our rebuttal, the statistics information of our dataset can be found in Appendix A, with the majority (i.e., training set) derived from [1] and others (i.e., validation and test set) derived from [6][7]. Here, as described in lines 650-654, the training set has 46,230 polymers, the validation set has 4,709 polymers, and the test set has 2,088 polymers.
> > >
> > >
> > > **4.2 PolyInfo is known to prohibit web scraping, so I am confused as to how was data obtained from this source?**
> > >
> > > As we have responded in our rebuttal, we only need polymer SMILES strings to run MD, and all these strings (including training, validation and test sets) we used are publicly available from previous works [1]-[5].
> > >
> > >
> > > **4.3 A dimensionality reduction method (e.g., UMAP) could help visualize the overlap in building block SMILES from each source, and would make for an interesting analysis to tell us how different the building blocks from each source are.**
> > >
> > > According to your suggestion, we have visualized the polymer SMILES strings from the training/validation/test set using the UMAP in this link (https://anonymous.4open.science/r/PolyConf/dataset/UMAP.png). Please note that our PolyConf is trained on the training set, while the best checkpoint is chosen based on the validation set.
> > >
> > > > **5. For MD validation (Q5. Validation of MD simulations):**
> > >
> > > As we have responded in our rebuttal, we invest significant effort to validate our MD simulations, including seeking guidance from experienced experts, examining the energy convergence, calculating the density values of typical polymers, and comparing them with experiment density values. The details can be found in our response to Reviewer 9jKE (https://openreview.net/forum?id=BsTLUx38qV&noteId=KDYQcy2SLb).
> > >
> > >
> > > **According to ICML 2025 Reviewer Instructions, the discussion between authors and reviewers is restricted to at most one additional round of back-and-forth, which means that we might no longer have the opportunity to respond to any further feedback from you. In summary, we have done our best to develop and refine our work, and answer all your questions. We hope the above responses can resolve your remaining concerns and enhance your confidence to increase the score. Thanks in advance for your consideration.**
> > >
> > > Regards,
> > >
> > > The authors of PolyConf
> > >
> > > [6] Polyinfo: Polymer database for polymeric materials design. EIDWT2011.
> > >
> > > [7] Transferring a molecular foundation model for polymer property predictions. Journal of Chemical Information and Modeling, 2023.

---

### Official Review · Reviewer_9jKE · 2025-03-13

**Overall Recommendation:** 3

**Summary:**

This paper introduces ​PolyConf, a novel hierarchical generative framework for polymer conformation generation. Addressing the unique challenges of polymers—such as high flexibility, large chemical space, and lack of prior datasets—PolyConf decomposes the task into two phases: Repeating Unit Conformation Generation and ​Orientation Transformation Generation

The authors also present ​PolyBench, the first benchmark dataset for polymer conformation generation, containing over 50,000 polymer conformations derived from molecular dynamics simulations.

**Claims And Evidence:**

**Strength**

- Superior Performance Over Baselines: The structural (S-MAT-R/P) and energy (E-MAT-R/P) metrics demonstrate PolyConf’s significant improvements over methods like TorsionalDiff (e.g., ​35.02 vs. 53.21 in S-MAT-R mean).
Results align with the hierarchical design’s intent to address polymer-specific challenges (flexibility, lack of rigid backbones).
​
- Efficiency: Timing comparisons (Figure 5) validate PolyConf’s speed (0.4 minutes vs. 3.54 minutes for GeoDiff).
​
- Scalability: Tests on doubled polymer sizes (4,000 atoms) show consistent performance (e.g., ​65.04 S-MAT-R mean vs. 119.29 for TorsionalDiff), supporting scalability claims.
​
- Dataset:  PolyBench’s size (50k+ conformations) and diversity (20–100+ repeating units) address the polymer data scarcity problem.
​

**Weakness**

- It would be beneficial for the paper to also test the proposed model on protein data, as proteins are a specific type of polymer. Given the abundance of available data and the well-established baselines, it is not essential for the proposed method to outperform existing folding models that leverage evolutionary information. However, it would be intriguing to assess where the method’s capabilities stand in comparison.

- It is unclear whether the dataset includes branched or cross-linked systems, or if it is limited to linear polymers only.
​
- PolyBench conformations derive from a single force field (AMBER). Force-field inaccuracies or parameterization biases may propagate into the dataset.
​Reproducibility:

- Missing Details: Training hyperparameters, computational resources, and force-field settings are not fully disclosed.
​
- While PolyBench is large, its quality hinges on force-field accuracy. The lack of experimental validation weakens claims about its representativeness.

- The core claims (performance, efficiency, scalability) are supported by internal benchmarks, but external validation (experimental data, broader baselines) is needed for robustness. The dataset’s reliance on simulations and omission of complex polymer types limit its universality. While PolyConf advances polymer informatics, claims about physical reliability and generalization require further evidence.

**Essential References Not Discussed:**

-

**Experimental Designs Or Analyses:**

Yes, but I don't identify critical issues.

**Methods And Evaluation Criteria:**

The methods and evaluation criteria are ​well-designed for the stated problem

**Other Comments Or Suggestions:**

-

**Other Strengths And Weaknesses:**

-

**Questions For Authors:**

**Q1:** ​Why was the method not tested on proteins, given the structural similarities between polymers and proteins ?

​**Q2:** Does the PolyBench dataset include branched/cross-linked polymers, or is it limited to linear chain polymers?

**Q3:** ​How was the quality of the PolyBench dataset validated beyond force-field-generated conformations?

**Relation To Broader Scientific Literature:**

-

**Theoretical Claims:**

There are no theorems to verify.

---

> ### Author Rebuttal · Authors · 2025-03-31
>
> Thanks for your comments. Below, we categorize and resolve your concerns.
>
> **W1&Q1: ​Why not test on proteins?**
>
> Our work focuses on polymers, aiming to explore polymer conformation generation (all-atom conformation).
> Polymers present unique challenges compared to proteins, such as greater structural flexibility (see lines 80-98 for a detailed explanation). In this context, our PolyConf is specifically designed to address and adapt to the distinct structural characteristics of polymers, making it difficult to apply it directly to proteins.
>
> In addition, while testing on proteins is out-of-the-scope of this work, we believe that PolyConf holds significant potential for application to other macromolecules composed of building blocks, such as proteins (amino acids) and RNA (nucleotides). Due to our limited experience with proteins and the time constraints, it is difficult for us to include them at this stage. We would, however, welcome the opportunity to collaborate with researchers in related fields to further explore and expand the potential applications of PolyConf.
>
> **W2&Q2: Details of PolyBench**
>
> As described in Appendix A.1, our molecular dynamics simulations are based on polymer SMILES strings. Since most publicly available polymer SMILES strings represent linear polymers, the current PolyBench dataset is also limited to linear polymers. Even though, to the best of our knowledge, PolyBench is the first benchmark for polymer conformation generation.
>
> In the future, we will continuously maintain and expand PolyBench to include a broader range of polymer data, especially branched and cross-linked polymers.
>
>
> **W3&W4: Reproducibility**
>
> The code, data, and various scripts for our PolyConf and molecular dynamics simulations are available in this anonymous link (https://anonymous.4open.science/r/PolyConf). We are confident that it has provided enough details to reproduce our work. Due to the repository size limitation, we can only provide a small subset of the complete dataset, but the complete dataset will be provided after acceptance.
>
> In particular, the training hyperparameters are provided in the corresponding training scripts, all experiments are implemented on eight A100 80G GPUs, and the force-field settings are provided in the `./MD/utils` folder of the provided anonymous repository. We will also explicitly state these details in the revised paper.
>
>
> **W3&W5&W6&Q3: Validation of MD simulations and Quality of PolyBench**
>
> We have invested significant effort to ensure the reliability of our molecular dynamics simulations, thereby guaranteeing the high-quality of the PolyBench dataset:
> * Under the guidance of highly experienced experts, we design our molecular dynamics simulations using standard pipelines widely adopted in previous works [1]. All scripts and settings related to the molecular dynamics simulations are available in the `./MD` folder of the provided anonymous repository, ensuring that our molecular dynamics simulations are fully transparent and reproducible.
> * We have analyzed the energy changes of various polymers during our molecular dynamics simulations, and the results show that most simulations achieve convergence within 1 ns, while we run the simulations for 5 ns to ensure the reliability and robustness of the PolyBench dataset.
> * We have calculated the density values of typical polymers through our molecular dynamics simulations and compared them with those experimental values provided in [1]. As shown in the following Table, the density values obtained via our molecular dynamics simulations are very close to those experimental values, further supporting the reliability of our molecular dynamics simulations and PolyBench dataset.
> | Polymer                     | Experiment density (g/cc) | MD density (g/cc) |
> |---|---|---|
> | [\*]CC(C)[\*]              | 0.850                    | 0.837             |
> | [\*]CC(C)O[\*]             | 1.125                    | 1.019             |
> | [\*]CC(Cl)(Cl)[\*]         | 1.630                    | 1.570             |
> | [\*]CCCCCCO[\*]            | 0.932                    | 0.944             |
> | [\*]CCCCCCCCO[\*]          | 0.906                    | 0.927             |
> * In addition, we provide some raw outputs of our molecular dynamics simulations (i.e., `NPT_Cases.tar.xz` and `NVT_Cases.tar.xz`)  in the `./MD` folder of the provided anonymous repository for validation.
>
> We hope the above responses can resolve your concerns. Since polymer conformation generation remains a relatively unexplored research area, we have invested significant time and effort into developing and refining our PolyConf and PolyBench to boost subsequent studies. Therefore, we kindly request your understanding of the challenges and complexities involved in pioneering work in this field.
>
> [1] High-throughput molecular dynamics simulations and validation of thermophysical properties of polymers for various applications. ACS Applied Polymer Materials, 2020.

---

### Decision · Program_Chairs · 2025-05-01

**Decision:**

Accept (poster)

**Comment:**

This paper proposes a dataset and method for polymer conformation generation. Overall, the reviewers are mildly enthusiastic about the work. The work is interesting in the sense that polymer is not a well studied problem in ML. On the other hand, the technical challenges are similar to those of other materials and molecules. The addition of a new benchmark is valuable given this is a new area. Thus I tend to recommend for accept, but would not be upset if it is not accepted.